# The Impact of Laser Thermal Effect on Histological Evaluation of Oral Soft Tissue Biopsy: Systematic Review

**DOI:** 10.3390/dj11020028

**Published:** 2023-01-18

**Authors:** Gianluca Tenore, Ahmed Mohsen, Alessandro Nuvoli, Gaspare Palaia, Federica Rocchetti, Cira Rosaria Tiziana Di Gioia, Andrea Cicconetti, Umberto Romeo, Alessandro Del Vecchio

**Affiliations:** 1Department of Oral Sciences and Maxillofacial Surgery, Sapienza University of Rome, 00161 Rome, Italy; 2Department of Radiological, Oncological and Pathological Sciences, Sapienza University of Rome, 00161 Rome, Italy

**Keywords:** CO_2_ laser, diode laser, Er:YAG, Nd:YAG, oral biopsy, thermal effect

## Abstract

The aim of the study is to review the literature to observe studies that evaluate the extent of the thermal effect of different laser wavelengths on the histological evaluation of oral soft tissue biopsies. An electronic search for published studies was performed on the PubMed and Scopus databases between July 2020 and November 2022. After the selection process, all the included studies were subjected to quality assessment and data extraction processes. A total of 28 studies met the eligibility criteria. The most studied laser was the carbon dioxide (CO_2_) laser, followed by the diode laser 940 nm–980 nm. Six studies were focused on each of the Erbium-doped Yttrium Aluminium Garnet (Er:YAG), Neodymium-doped Yttrium Aluminum Garnet (Nd:YAG) lasers, and diode lasers of 808 nm and 445 nm. Three studies were for the Potassium Titanyl Phosphate (KTP) laser, and four studies were for the Erbium, Chromium-doped Yttrium, Scandium, Gallium, and Garnet (Er,Cr:YSGG) laser. The quality and bias assessment revealed that almost all the animal studies were at a low risk of bias (RoB) in the considered domains of the used assessment tool except the allocation concealment domain in the selection bias and the blinding domain in the performance bias, where these domains were awarded an unclear or high score in almost all the included animal studies. For clinical studies, the range of the total RoB score in the comparative studies was 14 to 23, while in the non-comparative studies, it was 11 to 15. Almost all the studies concluded that the thermal effect of different laser wavelengths did not hinder the histological diagnosis. This literature review showed some observations. The thermal effect occurred with different wavelengths and parameters and what should be done is to minimize it by better adjusting the laser parameters. The extension of margins during the collection of laser oral biopsies and the use of laser only in non-suspicious lesions are recommended because of the difficulty of the histopathologist to assess the extension and grade of dysplasia at the surgical margins. The comparison of the thermal effect between different studies was impossible due to the presence of methodological heterogeneity.

## 1. Introduction

The oral biopsy is considered the diagnostic gold standard of oral soft tissue lesions, due to its ability to analyze lesions at the histopathological levels [1]. Several surgical cutting instruments can be used to perform oral biopsies such as a scalpel, electrosurgery, and lasers [2].

Lasers have been extendedly used in oral surgery throughout the past 30 years as an incision tool. Nowadays more than ten different laser devices are available for dental use [3]. They can be classified according to wavelengths, active medium, power, or the produced biological effects [4]. Enhancements and improvements to the clinical and surgical procedures have been reported with laser use when compared with the cold blade, including the high degree of decontamination of the surgical site, the minimal postoperative bleeding, and the significant reduction of inflammation and postoperative pain [5].

The interaction between the laser beam and the target tissue is the fundamental consideration for the clinical use and selection of laser devices. Each laser wavelength has an affinity to specific human tissue chromophores which can absorb specific radiations in the visible and invisible portions of the electromagnetic spectrum. These chromophores are melanin, water, hemoglobin, and hydroxyapatite.

The laser soft tissue cutting effect is based on the photothermal effect of laser, where the laser interacts with specific tissue chromophores leading to the heating and vaporization of the targeted tissues, and eventually the tissue dissection. This mechanism of action leads to a thermal effect at the margins of specimens [4,5].

Since safe and readable margins of collected samples, in particular of suspicious dysplastic or neoplastic lesions, are extremely important, several studies have been conducted with different lasers and parameters to evaluate the extent of the thermal effect and its influence on the histological evaluation and interpretation, in order to achieve a standardized protocol exploiting the maximum benefits and the minimal drawbacks of the laser as an incision tool [3,5,6].

To our knowledge, the literature might have not been reviewed systematically to observe studies investigating this issue. The aim of the study is to review the literature to observe studies assessing the thermal effect with different laser wavelengths on the histological evaluation of oral soft tissue biopsies and to analyze the difference in the resulting thermal effects in soft tissue specimens collected by different wavelengths and parameters. The laser wavelengths considered in this review are the Potassium Titanyl Phosphate (KTP) laser of 532 nm; diode lasers of 445 nm, 808 nm, and 940 nm–980 nm; Neodymium-doped Yttrium Aluminum Garnet (Nd:YAG) laser of 1064 nm; Erbium, Chromium-doped Yttrium, Scandium, Gallium and Garnet (Er,Cr:YSGG) laser of 2780 nm; Erbium-doped Yttrium Aluminium Garnet (Er:YAG) laser of 2940 nm; and carbon dioxide (CO_2_) laser.

## 2. Materials and Methods

A systematic literature review was conducted following the parameters of PRISMA, “Preferred Reporting Items for Systematic Reviews and Meta-analyses”, guidelines.

The focus question was: “what is the impact of the thermal effect of different laser wavelengths on the histological evaluation of laser-collected oral biopsy?”.

The systematic review was registered at PROSPERO, “the International Prospective Register of Systematic Reviews”, with registration nr. CRD42022385059.

### 2.1. Eligibility Criteria

The inclusion criteria of the studies on this systematic review were studies that evaluated the thermal effect of lasers on soft tissue biopsies specimens, reporting the measurements of the thermal effect histologically, utilizing one of the following wavelengths or all of them: 532 nm KTP; diode lasers of 445 nm, 808 nm, and 980 nm; 1064 nm Nd:YAG laser; 2780 nm Er,Cr:YSGG laser; 2940 nm Er:YAG laser; and CO_2_ laser. All the studies had to be written in the English language. Almost all kinds of studies were considered: Randomized Controlled Trials (RCT), clinical trials, and animal (in vivo or ex vivo) studies.

Studies with incomplete experimental data, reviews (narrative and/or systematic), case reports, abstracts, and letters to editors were excluded from this review.

### 2.2. Search Strategy

The PubMed and Scopus databases were searched between July 2020 and November 2022. All MeSH terms, keywords, and terms related to laser, oral, biopsy, thermal, and effect were used and combined with Boolean operators “AND” and “OR” (Table 1). A manual search was also performed on the citation and reference lists of the included studies to identify other publications not recalled in the initial databases search.

### 2.3. Study Selection

The studies collected from the database search were screened independently in two stages by two reviewers (A.M. and A.N.). An independent screening of the titles and abstracts of the resulting studies was performed in the first stage based on the aforementioned inclusion and exclusion criteria. In the second stage, a full-text read of the selected studies was performed to confirm the suitability of the articles for the review. An arbitration and discussion with a third reviewer (G.T.) were conducted in case of disagreements between the two reviewers at both stages.

### 2.4. Extraction and Synthesis of Data

The collection and synthesis of data were performed from each selected eligible study by the same two reviewers. The extracted data were: the author/year, study type (animal or clinical), laser wavelength, specimen type (including the clinical and histopathological perspectives), number of specimens (only specimens collected on oral soft tissues by the considered laser wavelengths in this revision), employed microscope, magnification power, method of histological evaluation, number of pathologists, main outcomes, and conclusions. In addition, the reported thermal effects were extracted from each study. The laser parameters were also collected including the wavelength (nm), type of emitter, groups (if present), number of samples (*n*) for each group and for each wavelength if present, frequency (Hz), energy (mJ), air/water ratio, fiber/spot diameter (μm), fluence (J/cm^2^), irradiance (W/cm^2^), and power (W). Conversion of units was performed for laser parameters and thermal effects if needed in order to have a better homogeneousness of the presented data.

The resulting data for all the included studies were synthesized tabularly and divided into (A) data that demonstrate the general overview of the sample distribution and evaluation details, and (B) data that demonstrate the laser parameters and the resulting thermal effects. At this stage, the third reviewer (G.T.) was consulted in case of any disagreements between the two reviewers (A.M. and A.N.).

Heterogeneousness of the data was observed among the included studies concerning the study design, laser parameters, sample type, histological evaluation methods, and reported thermal effect that hindered the authors from carrying out the meta-analysis. Therefore, only a narrative review and tabular synthesis of data were performed.

### 2.5. Assessment of Quality and Bias

For the assessment of the quality and risk of bias (RoB) of the included studies, different assessment tools were employed according to the type of the studies. In animal studies, SYRCLE’s (“Systematic Review Centre for Laboratory Animal Experimentation”) tool was used [7]. This tool consists of six types of bias with 10 domains. Each domain is scored with low, unclear, or high risk.

In clinical studies, the MINORS (“Methodological Index for Non-Randomized Studies”) tool was used [8]. This quality assessment tool consists of 12 methodological items. The first eight items can be applied to both comparative and non-comparative studies, while the remaining four items are applied only to comparative studies. The scores are “0” for not reported, “1” for reported but inadequate, or “2” for reported and adequate. At the end of the assessment, the total score should be calculated where the ideal score would be 16 for non-comparative studies and 24 for comparative studies.

## 3. Results

The initial search on the PubMed database resulted in a total of 119 studies in the time period between 1987 and 2022. On the Scopus database, 1433 studies were identified in the time period between 1990 and 2022. After the removal of duplicates and the removal of studies by the automated tools, a total of 152 studies were subjected to the title and abstract screening. After the second stage of the screening (full-text read), a total of 23 studies met the eligibility criteria. Another 5 studies were identified from the reference and citation lists. A total of 28 studies were included in this review and subjected to the extraction of data and quality assessment (Figure 1) [9].

The exclusion of the 28 studies was due to different reasons, including studies that use different wavelengths than considered ones, studies that use only qualitative methods for the description of thermal effects, studies with incomplete experimental data, and/or studies with an incomplete report of important parameters.

The included studies were distributed as follows: fourteen animal studies and 14 clinical studies. The qualitative analysis was performed for all the included studies using different RoB assessment tools (SYRCLE and MINORS) according to the kind of study. The allocation concealment domain in the selection bias and the blinding domain in the performance bias were awarded unclear or high in almost all the included animal studies. For clinical studies, there were six non-comparative studies and eight comparative studies. The range of the total RoB score in the comparative studies was 14 to 23, while in the non-comparative studies, it was 11 to 15. Figure 2 and Figure 3 show the scores of different considered domains of the used RoB assessment tools of all the included studies.

### 3.1. Animal Studies

A total of 607 samples were collected in the animal studies for the histological evaluation of the thermal effect of lasers. Table 2 shows an overview and evaluation details of all included animal studies. The samples were harvested from pig tongue [5,10,11,12,13,14,15,16], bovine tongue [17], pig oral mucosa [18], rabbit ventral surface of the tongue [19], Sprague rat tongue [20], and porcine oral mucosa and gingiva [3,21]. The examined wavelengths were distributed among the animal studies as follows: 808 nm laser in three studies [10,17,21], 940 nm–980 nm laser in five studies [10,13,15,16,18], 445 nm laser in three studies [14,17,18], Nd:YAG laser in four studies [10,13,17,21], Er,Cr:YSGG laser in four studies [3,10,19,21], KTP laser in two studies [11,17], Er:YAG laser in three studies [12,13,21], and CO_2_ laser in five studies [3,5,13,20,21].

The histological evaluation was conducted in all the included animal studies by optical microscopes with a magnification range ranging from 10× to 40×. In six studies, the histological evaluation was made by two pathologists, while in five studies it was made by only one pathologist. Moreover, this issue was not clearly declared in the other three studies.

Different methods were considered for the evaluation of the thermal effects of lasers. Only qualitative evaluations were carried out in five studies [10,11,12,17,19]. Rizoiu et al. [19] and Romeo et al. [10] based their evaluation on identifying the artefactual changes, the marginal coagulation zones of hyalinization, and the degree of inflammation. In two studies by Romeo et al. [11,12], the evaluation of specimens was through identifying the Thermal Damage Score (TDS) that consists of a scale of four items from 0 to 3 (0 “no damage”, 1 “little damage”, 2 “moderate damage”, and 3 “severe damage”), while Fornaini et al. [17] evaluated the cut quality by assigning a score to the specimens from 0 to 5, where 5 was the highest quality with the cold blade.

The other nine studies performed both qualitative and quantitative evaluations of the specimens [3,5,13,14,15,16,18,20,21]. Palaia et al. [5,14,15] in their studies evaluated the specimens identifying the TDS and determining the width of peripheral thermal effects. Azevedo et al. [13] evaluated the specimens macroscopically based on the criteria of Cercadillo-Ibarguren (a scale of 0 to 4) and histologically through identifying the epithelial, Connective Tissue (CT), vascular changes, incision morphology, and the overall width of tissue modifications [3]. Braun et al. [18] measured the Maximum Denaturation Depth (MDD) and Denaturation Area (DA) for all the included specimens. Kawamura et al. [21] made the histological and histometric analysis of the coagulated and thermally affected layers at the ablation bottom. Cercadillo-Ibarguren et al. [3] performed macroscopic and histological evaluation through the measurement of the extension of the hyalinized or coagulated tissue adjacent to the irradiation edge. Seoane et al. [20] evaluated histologically the epithelial features, the number of artefacts, and the width of thermal damage. Prado et al. [16] calculated histologically the Thermal Damage Depth (TDD) and the Thermal Damage Area (TDA), where they calculated the TDA by calculating the difference between the total area of each specimen and its total area of thermal damage.

### 3.2. Clinical Studies

A total of 1117 specimens were collected in the clinical studies and the evaluation of the thermal effect of the used lasers was performed. All the specimens were collected for the diagnosis of oral soft tissue lesions. The histopathological diagnosis was reported for a total of 941 specimens, while 176 specimens were not clearly identified in four clinical studies. Figure 4 shows the distribution of the specimens according to the diagnosis of lesions. The overview and evaluation details of all included clinical studies are resumed in Table 3.

In three studies, the specimens were limited to fibrous hyperplasia lesions [22,23,24]. In five studies, the localization of the oral soft tissue lesions was considered as an inclusion criterion. It was the tongue in one study [25], the buccal mucosa in three studies [22,23,24], and both cheek and buccal mucosa in one study [26]. There were six non-comparative studies [22,27,28,29,30,31], and eight comparative studies [23,24,25,26,32,33,34,35]. Among the included studies, there were two retrospective studies [28,34] and two RCTs [22,24].

The distribution of the examined wavelengths among the clinical studies was the following: 808 nm in three studies [28,32,35], 980 nm in three studies [33,34,35], 445 nm in three studies [30,31,33], Nd:YAG laser in two studies [26,34], KTP laser in one study [32], Er:YAG laser in three studies [23,24,34], and CO_2_ laser in seven studies [22,23,24,25,27,29,34].

In the clinical studies, the optical microscope was mentioned in the methods for the histological evaluation in all the studies except for four studies [22,23,24,33]. The magnification range was from 5× to 100×. One pathologist was responsible for the histological evaluation in ten studies [22,23,24,26,29,30,31,32,33,34], and two pathologists in two studies [28,35]. In two studies, the number of examiners was not precise [25,27].

The thermal effect of the examined wavelengths was evaluated qualitatively and quantitively in all the clinical studies by different methods. The epithelial, CT, vascular changes, incision morphology, and overall width of tissue modifications were considered in three studies [26,28,34]. In three studies by Suter et al. [22,23,24] and a study by Gill et al. [35], the maximum, minimum, mean, and median values of the Thermal Damage Zone (TDZ) were measured. Matsumoto et al. [25] measured the distances from the edges of the specimen to the thermal artifacts. In four studies, the thermal effect at the peri-incisional margins was measured quantitively for both the epithelium and CT [29,30,31,32]. In one study, the tissue necrosis width was measured for five different soft tissues (epithelium, connective tissue “loose and dense”, muscle, and salivary gland epithelium) [27]. Gobbo et al. [33] quantified the maximum thermal damage along the cutting margin.

### 3.3. Laser Parameters

There was a difference in the examined parameters for each considered laser system. Among the considered wavelengths, the most evaluated laser system for the thermal effect on soft tissues was the CO_2_ laser followed by the diode lasers 940 nm–980 nm. The least evaluated was the KTP laser in only three studies [11,17,32]. The examined laser parameters of all the included studies were summarized in Table 4 and distributed according to the wavelength and accompanied by the reported thermal effect.

## 4. Discussion

Despite the differences and heterogeneity among the included studies, almost all of them showed that the different tested laser wavelengths did not have a thermal effect that may hinder the histological diagnosis. It appears that the use of laser in oral biopsy should be tailored by managing variables related to the extent of histological thermal effects, including the histological condition of the target tissue (healthy or pathological), the size of the excised lesion, the operator experience, and the source of energy emission, to minimize the thermal effect that cannot be prevented.

Concerning the 808 nm laser, six studies (three clinical and three animal studies) evaluated the thermal effect on soft tissues [10,17,21,28,32,35]. The range of the tested powers was 1 to 3 W. The fluence was reported in two studies and ranged from 284 to 2400 J/cm^2^ [10,32]. The irradiance was reported only in two studies with a range of 1415.4 to 2400 W/cm^2^ [10,21]. The spot size was 320 µm in all the studies except for one study in which it was 300 µm [21]. The laser was tested in Continuous Wave (CW) mode in all the studies except for one study where it was tested in Pulsed Wave (PW) with a Ton of 100 ms and Toff of 100 ms [10]. The range of the reported thermal effect in clinical studies with almost similar parameters was 17.92 to 473 µm [28,32]. All the studies suggested that the 808 nm laser can be considered an affordable tool and confirm the possibility of having a clear histological diagnosis. However, the extension of margins during the excision of oral lesions was also recommended [10,32]. In one study, it was recommended to use this laser on oral lesions with a diameter of more than 3 mm because they found that specimens under 3 mm had frequently significant epithelial, stromal, and/or vascular changes and the diagnosis in 46.15% of these specimens was not achievable [28].

Among the eight studies of diode laser 940–980 nm, five studies (two animal and three clinical) reported the extent of the thermal effect with a range of 100 to 1198.54 µm [13,15,33,34,35]. Two of them tested the power of 3.5 W in PW and one at 2 W in PW. In one study, the MDD and the DA were evaluated with a power of 3 W in CW and PW, where the mean MDD was 208 μm in CW and 124 μm in PW, and the mean DA was 127 μm in CW and 307 μm in PW [18]. Among the eight studies, three studies reported fluence. The range of the tested fluence was 99.2 to 4777 J/cm^2^. The irradiance was reported in three studies and ranged from 1415.4 to 4957.5 W/cm^2^. The range of tested powers was 1 to 6 W. The reported spot size was 320 μm in three studies, 300 μm in two studies, 400 μm in one study, and not mentioned in two studies. All the studies stated that diode laser 940–980 nm laser did not hinder the histological evaluation, and two studies recommended the use of laser in pulsed mode with the extension of margins [10,18].

Concerning the 445 nm laser (Blue laser), all the included six studies (three animal and three clinical) reported the best quality of cut and minimal thermal effect in comparison with others [14,17,18,31,33,35]. In three studies (one animal and two clinical), the extent of thermal effect was reported for the epithelium and CT [14,30,31]. The mean extent of the thermal effect with almost the same parameters was evidently higher on the epithelium in the clinical study than that in the animal study, whereas the mean extent of thermal effect on the epithelium, with approximately 2 W in CW, was 137.5 μm in the animal study, and 650.93 μm in the clinical study, while for the CT, the mean extent of thermal effect with the same abovementioned parameters was approximately similar between the clinical and animal studies [14,30,31]. All the studies on the 445 nm laser used a fiber of 320 μm and tested the range of power from 1.4 to 4 W. Only two studies reported fluence (3100 J/cm^2^) [30,31].

Six included studies (four animal and two clinical studies) evaluated histologically the thermal effect of the Nd:YAG laser on the excision of soft tissue lesions [10,13,17,21,26,34]. All of them tested the laser in PW. The fiber size was 300 μm in three studies [13,21,34], 320 μm in two studies [17,26], and 400 μm in one study [10]. The range of tested powers was from 1 to 6 W. The fluence was reported in three studies with a range of 95.5 to 141.6 J/cm^2^ [10,21,34], and the irradiance was reported in three studies with a range of 300 to 5665.7 W/cm^2^ [4,6,23]. The total extent of the thermal effect was measured in four studies [13,21,26,34]. Among them, two clinical studies evaluated the thermal effect on the epithelium and CT [13,34]. The thermal effect on CT was quite similar in both studies when comparing the results of approximately similar parameters (3.5 W in PW and 4 W in PW), where it was 376.6 μm and 310.85 μm, respectively, while the thermal effect on epithelium was clearly different between the two studies with these above-mentioned parameters, where it was 305.8 μm and 899.83 μm, respectively [13,34]. In all the included studies, the extent of the thermal effect with Nd:YAG was the highest when compared to other lasers and, in one clinical study by Vescovi et al., serious thermal effects were observed in small specimens (less than 7 mm) independently from the settings employed [26].

Er,Cr:YSGG laser was tested in four animal studies [3,10,19,21]. The spot size was 600 μm in three studies and 680 μm in one study. The irradiance was reported only in one study with a range of 707 to 1000 W/cm^2^ and the fluence was reported in two studies with a range of 35 to 53 J/cm^2^ [10,21]. The reported extent of thermal effect was quite small with all tested powers in particular with the use of air/water spray. In two studies with similar parameters, the range of total extent of thermal effect was 9.26 to 33.1 μm. The parameters were a power of 1 W in PW (20 Hz) with air/water spray and a spot diameter of 600 μm [3,21].

All the studies that tested the KTP laser demonstrated a low thermal effect on soft tissues [11,17,32]. The range of tested powers was 1.5 to 3 W. Only one clinical study reported the extent of the thermal effect with a range of 196 to 213 μm with a power of 1.5 W in PW and a fiber diameter of 300 μm [32].

Er:YAG laser was tested in six studies (three animal and three clinical studies) and almost all of them showed that it is a conservative tool [12,13,21,23,24,34]. All the included studies tested the Er:YAG laser in PW mode. The range of tested powers was 1 to 7 W. In two animal studies, Er:YAG laser was tested with and without air/water spray in PW with different powers [13,21]. Both studies demonstrated a lower thermal effect with the use of air/water spray. In all the studies that compared the Er:YAG laser with other laser systems, the Er:YAG laser showed the least thermal effect [13,21,23,24,34].

The CO_2_ laser was the most studied laser: it was tested in twelve studies (five animal and seven clinical studies) [3,5,13,20,21,22,23,24,25,27,29,34]. The range of tested powers was 1 to 20 W. The fluence was reported in only one study (40.8 J/cm^2^) [34]. The irradiance was reported in three studies (796.2, 2040.8, and 2320 W/cm^2^) [21,27,34]. All the studies confirmed that it is a reliable tool for the excision of soft tissues. CO_2_ laser showed the lowest thermal effect when tested in studies that compare it with other laser systems except for Er:YAG and Er,Cr:YSGG lasers [3,13,21,23,24,34]. In one study, the authors reported produced epithelial damage for both low- and high-tested powers (3 W to 12 W) similar to light dysplasia features that could cause erroneous therapy. Interestingly, the lowest measured thermal effect was observed in this study in the group of high power (12 W) [20].

From this revision, some observations must be highlighted. First, the reported thermal effects were slightly different among animal studies and clinical studies, which is predictable because animal studies were usually done in ex vivo conditions, where the specimens were deprived of blood when compared to clinical conditions mainly for wavelengths highly absorbed by Haemoglobin (Hb). Second, even similar studies showed differences in the extent of the thermal effect that may be influenced by the difference in effects on pathologic tissue depending on the lesion characteristics. Third, although almost all the studies conclude that the thermal effect of different laser wavelengths does not hinder the histological diagnosis and evaluation, thermal damage occurs and what should be considered is to minimize the thermal effect by better adjusting laser parameters.

Finally, there were some limitations that need to be highlighted. First, the meta-analysis and comparison among the included studies were not applicable due to the diversity in the methodological approaches regarding the histometric analysis, laser parameters, and sample types. Second, a selection bias should be considered, as this systematic approach depended only on papers in the English language, and that were recruited by research on PubMed and Scopus databases. Third, some studies reported the extent of the thermal effect of the tested laser in a confusing way that hindered the comparison process. For future studies, it should be more advisable to include the micrometric measurements of both epithelial and CT thermal effects.

## 5. Conclusions

From this literature review, there emerged some observations. Some studies recommended the extension of margins during the collection of laser oral biopsies and some of them recommended the use of laser only in non-suspicious lesions because of the possible difficulty of the histopathologist to assess the extension and grade of dysplasia at the surgical margins. The comparison of the thermal effect between different studies was impossible due to the presence of methodological heterogeneity. Further studies are needed with a standardized approach with micrometric measurements of epithelial and connective tissue thermal effects for comparative purposes and eventually determining a standard approach and parameters for the use of each type of laser on oral biopsies, in particular because there are many reported advantages to using lasers that may eventually improve patients’ management, such as the lesser use of local anaesthesia, the anti-inflammatory effect, and the possibility to have a bloodless operation field.

## Figures and Tables

**Figure 1 dentistry-11-00028-f001:**
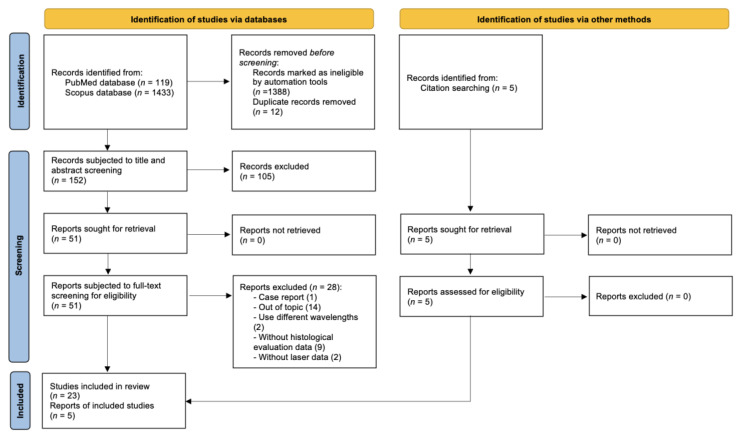
Flow diagram shows the study selection process.

**Figure 2 dentistry-11-00028-f002:**
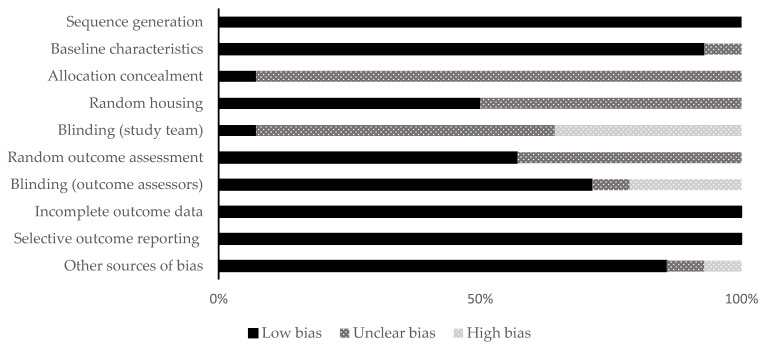
Quality and risk of bias assessment scores of the animal studies using the SYRCLE’s tool.

**Figure 3 dentistry-11-00028-f003:**
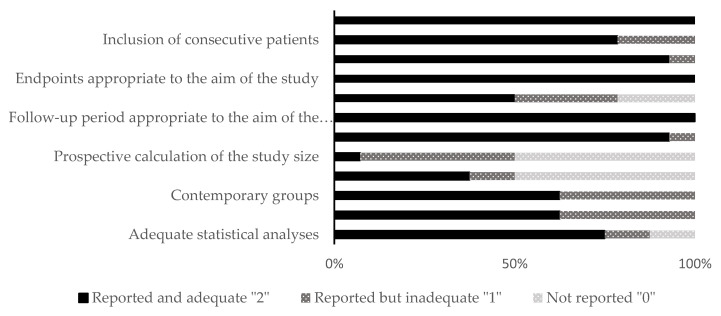
Quality and risk of bias assessment scores of clinical studies (in vivo) using the MINORS tool.

**Figure 4 dentistry-11-00028-f004:**
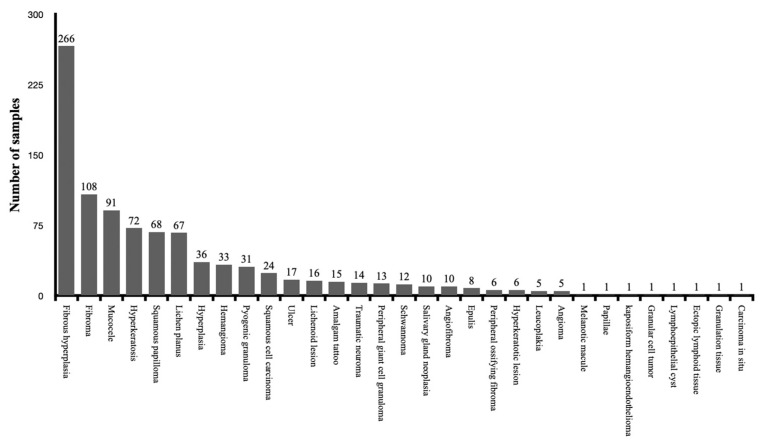
The distribution of the specimens according to the diagnosis of lesions among the clinical studies.

**Table 1 dentistry-11-00028-t001:** Shows the used search strategy in the PubMed and Scopus databases.

Databases	Search Strategy
PubMed	{“laser s”(All Fields) OR “lasers”(MeSH Terms)OR “lasers”(All Fields) OR “laser”(All Fields) OR “lasered”(All Fields) OR “lasering”(All Fields)} AND {“mouth”(MeSH Terms) OR “mouth”(All Fields) OR “oral”(All Fields)} AND {“biopsie”(All Fields) OR “biopsy”(MeSH Terms) OR “biopsy”(All Fields) OR “biopsied”(All Fields) OR “biopsies”(All Fields) OR “biopsy s”(All Fields) OR “biopsying”(All Fields) OR “biopsys”(All Fields) OR “pathology”(MeSH Subheading) OR “pathology”(All Fields)} AND {“thermal”(All Fields) OR “thermalization”(All Fields) OR “thermalize”(All Fields) OR “thermalized”(All Fields) OR “thermalizes”(All Fields) OR “thermalizing”(All Fields) OR “thermally”(All Fields) OR “thermals”(All Fields)} AND {“effect”(All Fields) OR “effecting”(All Fields) OR “effective”(All Fields) OR “effectively”(All Fields) OR “effectiveness”(All Fields) OR “effectivenesses”(All Fields) OR “effectives”(All Fields) OR “effectivities”(All Fields) OR “effectivity”(All Fields) OR “effects”(All Fields)}
Scopus	laser AND oral AND biopsy AND thermal AND effect AND {LIMIT-TO (SUBJAREA, “DENT”)} AND {LIMIT-TO (DOCTYPE, “ar”)} AND {LIMIT-TO (LANGUAGE, “English”)}

**Table 2 dentistry-11-00028-t002:** Overview and evaluation details of all included animal studies.

Author, Year	Laser	Sample	Sample (*n*)	Type of Microscope	Magnification	Method of Evaluation	Nr. of Pathologists	Main Outcomes	Conclusions
1. Rizoiu et al., 1996 [19]	Er,Cr:YSGG	Ventral tongue mucosa of New Zealand white rabbits	1	Optical microscope	13×	Histological evaluation of artifactual changes, marginal coagulation zones of hyalinization, and degree of inflammation.	-	- Minimal edge coagulation and carbonization artifacts were observed. - No vacuolization changes on contiguous epithelial or connective tissues.	This laser can be used for the collection of diagnostic biopsy specimens.
2. Romeo et al., 2007 [10]	Diode lasers (808 nm, 980 nm), Nd:YAG, Er,Cr:YSGG	Pig tongues	36	Optical microscope	10× or 25×	Histological evaluation and determination of the exact extent of peripheral thermal damage and comparing among different lasers.	2	- The 808 nm in PW ^1^ and the Er,Cr:YSGG showed the best results. - Deeper thermal effect in the CT ^2^ with Nd:YAG even if the laser worked with lower fluence than diode lasers in CW ^3^.	These lasers can be employed in the biopsy investigations of dysplastic lesions with the extension of margins to ensure a safe histological evaluation.
3. Cercadillo-Ibarguren et al., 2010 [3]	Er,Cr:YSGG, CO_2_	Porcine oral mucosa	90	Optical microscope	40×	Macroscopic and histological evaluation through the measurement of the extension of the hyalinized or coagulated tissue adjacent to the irradiation edge.	1	- A wide range of thermal damage was observed with significant differences among different lasers and parameters. - The lowest thermal effect was observed with Er,Cr:YSGG laser using water/air spray followed by CO_2_ laser.	- The extent of thermal effect on soft tissues may be determined by the emission parameters of each laser type. - The wavelength remains the determinant of the absorption rate characteristics of every tissue and the thermal effect.
4. Romeo et al., 2010 [11]	KTP	Pig cadaver tongue	45	Optical microscope	10×	Histological analysis was performed by assigning a score from 0 to 3 to the peripheral thermal damage according to the following scale: 0—no damage, 1—little damage, 2—moderate damage, and 3—severe damage.	2	- The cut edges of the incision of all the specimens were free from histological artefacts. - No statistical differences were observed among the groups with a slight increase of thermal effect on groups with the higher power.	KTP laser showed surgical effectiveness and caused little peripheral damage to the cut edges.
5. Seoane et al., 2010 [20]	CO_2_	Sprague rat tongue	20	Optical microscope	20× and 40×	Histological evaluation included epithelial features, number of artefacts, and width of thermal damage.	2	- The histological artefacts were mainly localized in the basal and suprabasal layers of the oral epithelium. - No significant difference in the number and width of thermal damage was observed among the experimental groups (with low and high powers).	The epithelial damage produced by CO_2_ laser (low and high powers) was similar to light dysplasia features that may cause erroneous therapy.
6. Romeo et al., 2012 [12]	Er:YAG	Swine cadaver tongues	45	Optical microscope	10×	Histological analysis by assigning a score from 0 to 3 to each sample indicating the degree of peripheral thermal damage (TDS) and obtaining the average TDS ^4^.	2	- Less thermal damage was observed with intermediate parameters (at 80 and 100 mJ and 28 and 35 J/cm^2^). - No difference in the peripheral thermal damage was observed among the groups and the histological evaluation was possible in all the specimens.	Er:YAG laser can be safely employed for oral biopsy investigations with a successful histological evaluation.
7. Palaia et al., 2014 [5]	CO_2_	Pig cadaver tongue	30	Optical microscope	40×	Histological evaluation through assessing the TDS (from 0 to 3), and determining the width of peripheral thermal effects.	1	In all the samples, the histological readability was optimal and the thermal damage was negligible.	CO_2_ laser showed surgical effectiveness with little peripheral thermal damage, allowing a safe histological diagnosis.
8. Azevedo et al., 2016 [13]	Diode laser 980 nm, Nd:YAG, Er:YAG, CO_2_	Pig cadaver tongue	100	Optical microscope	-	- Macroscopic evaluation based on the criteria of Cercadillo-Ibarguren (a scale of 0 to 4). - Histologic evaluation including epithelial, CT, vascular changes, incision morphology, and overall width of tissue modifications.	2	- The lasers can be ordered according to the resulted thermal tissue damage extension as follows (from the greatest to the lowest): Nd:YAG laser, diode 980 nm (3.5 W and boost PW), CO_2_ laser (7 W in CW), CO_2_ laser (7 W in PW), diode 980 nm (3.5 W in PW), CO_2_ laser (3.5 W in PW), and Er:YAG laser. - The presence of an association between the tissue damage extension and the degree of carbonization was observed, and an association between the tissue damage extension and regularity of the incision.	- These lasers can be employed safely for oral biopsies with a successful histological analysis with taking into consideration the biological effects of each laser type. - Er:YAG laser and CO_2_ laser at 3.5 W in pulsed mode presented the best lasers for ensuring the successful histological evaluation.
9. Fornaini et al., 2016 [17]	Diode lasers (808 nm, 450 nm), Nd:YAG, KTP	Dorsal surface of bovine tongue	4	Low- and high-power microscope	-	Histological evaluation through assigning a cut quality score (from 0 to 5), where “5” is the highest with cold blade.	1	- The best quality of cut was observed with 450 nm laser. - The cut quality score for 450 nm and KTP was 3, 2 for 808 nm laser, and 1 for Nd:YAG.	Based on the results of this study, blue diode laser (450 nm) is suggested to be used in oral surgery in daily practice.
10. Braun et al., 2018 [18]	Diode lasers 970 nm, 445 nm	Pig cadavers oral mucosa	30	Optical microscope at 35-fold magnification	-	Histological evaluation of tissue denaturation including the measurement of MDD ^5^ and DA ^6^.	-	- The highest mean values of MDD were observed with the 970 nm laser in CW (with significance). - No significant difference was observed among different lasers in regard to the DAs. - The highest incision depth with the lowest amount of soft tissue denaturation was observed in the group of the 445 nm laser in contact mode (with statistical significance).	The 445 nm laser shows a higher cutting efficiency when compared to the 970 nm laser.
11. Kawamura et al., 2019 [21]	Diode laser 808 nm, Nd:YAG, Er,Cr:YSGG, Er:YAG, CO_2_	Porcine gingiva	56 (8 samples for each treatment set)	Optical microscope	-	Histological and histometric analysis of the coagulated layers and thermally affected layers at the ablation bottom.	2	- Sharp and deep grooves were detected with Er:YAG, Er,Cr:YSGG, and CO_2_, while the grooves were flatter with diode and Nd:YAG. - The minimal thickness of the coagulated layer and thermally affected layer were observed with Er:YAG and Er,Cr:YSGG. Then, they were followed by CO_2_, diode, and Nd:YAG, respectively. - The use of water spray with Er:YAG and Er,Cr:YSGG reduced significantly the coagulated layer and prevented the thermal effect for both.	Er:YAG laser demonstrated the most efficient tool among the other lasers for gingival ablation with minimal thermal effect in particular with the use of water spray.
12. Palaia et al., 2020 [14]	Blue diode (445 nm)	Pig cadaver tongues	30	Optical microscope	10× and 40×	Histological evaluation including the measurement of the width of thermal damage in the peri-incisional epithelium and CT.	1	- The thermal effect on epithelium was lower than on the CT in all the groups with significance. - The mean thermal effect in groups of PW was significantly lower than that of CW. - Evaluating groups with the same parameters and groups that differ in the mode of transmission (CW and PW), some of them showed significant difference toward the PW.	The 445 nm diode laser is suggested for biopsy investigations with excellent surgical properties.
13. Palaia et al., 2021 [15]	Diode laser 976 nm	Pig cadaver tongues	30	Optical microscope	5× and 10×.	Histological evaluation including the measurement of the width of thermal damage in the peri-incisional epithelium and CT.	1	- Readable cut margins were shown in all the samples with a small damaged area. - The damage extension was always smaller in the epithelium than in CT. - The highest epithelium damage was 689.6 μm in a sample taken at 6 W in PW, while the highest CT damage was 964.24 μm in a sample taken at 5 W in CW.	The 976 nm diode laser can be considered as a good surgical tool and allowed a safe histological diagnosis.
14. Prado et al., 2022 [16]	Diode laser 940 nm	Fresh pig tongues	90	Optical microscope	40×	Histological assessment of quantitative measures of thermal damage through calculating the thermal damage depth and the thermal damage area (which is the difference between the total area of each specimen and its total area of damage).	-	- No statistical differences in thermal damage depth were observed between different laser groups. - Only a statistical difference was observed between group 3 and other 2 groups at the level of the thermal damage area.	- The cutting efficiency of the super pulsed diode laser is comparable to traditional blade. - Appropriate parameters of this laser can produce surgical outcomes with less collateral damage.

^1^ Pulsed Wave (PW); ^2^ Connective Tissue (CT); ^3^ Continuous Wave (CW); ^4^ Thermal Damage Score (TDS); ^5^ Maximum Denaturation Depth (MDD); ^6^ Denaturation Area (DA).

**Table 3 dentistry-11-00028-t003:** Overview and evaluation details of all included clinical studies.

Author, Year	Laser	Pathological Perspectives		Sample (*n*)	Type of Microscope	Magnification	Method of Evaluation	Nr. of Pathologists	Main Outcomes	Conclusions
		Clinical	Histopathological (*n*)							
1. Pogrel et al., 1990 [27]	CO_2_	Oral soft tissue lesions	-	23	Optical microscope	100×	Histological evaluation through measuring the mean of tissue necrosis in microns for five different soft tissues (epithelium, connective tissue “loose and dense”, muscle, and salivary gland epithelium).	-	- The width of tissue necrosis was different among the different examined tissues types, that were probably based on the water content of each tissue type. - The dense fibrous tissue, mucosal epithelium, and muscle showed the widest tissue necrosis.	The relative narrow width of tissue necrosis with this tool and protocol may provide a superior properties on the wound healing when compared to those created by electrosurgery.
2. Matsumoto et al., 2008 [25]	CO_2_	Excised different tongue lesions, including malignant, precancerous, and benign lesions	SCC ^1^ (4), oral leucoplakia (5), fibroma (5), mucocele (4), granuloma (1), hemangioma (1)	20	Optical microscope	40× or 100×	Assessment and comparison histologically of the resulted artifactual patterns of thermal injury by the CO_2_ in (PW ^2^ and CW ^3^) laser and the electrotome. In addition, comparing the distances from the borders of the biopsy specimen to the thermal artifacts produced by each method.	-	- Thermal denaturation including carbonization, vacuolar degeneration, and elongation of nuclei, were reported at the peripheral margins for all the methods. - CO_2_ laser in PW reduced the amount of thermal denaturation.	CO_2_ laser, particularly in PW, was better than the electrotome.
3. Vescovi et al., 2010 [26]	Nd:YAG	Excised benign exophytic oral lesions (cheek and buccal mucosa)	Hyperplastic fibro-epithelial lesions (denture-induced fibrous hyperplasia, fibroma, and fibro-papilloma)	15	Low- and high power light microscope	40× and 100×	Histological evaluation including epithelial, CT ^4^, vascular changes, incision morphology, and overall width of tissue modifications.	1	- No significant difference with regard to stromal changes, vascular stasis, and regularity of incision was observed between specimens removed with two different parameters of Nd:YAG laser. - Specimens with a mean size less than 7 mm showed significant Epithelial and stromal changes. - Serious thermal effects in small specimens (mean size less than 7 mm) were observed independently from the frequency and power employed. - Specimens obtained with higher frequency and lower power showed better quality of incision and less width of overall tissue injuries.	- Nd:YAG laser may cause serious thermal effects in small specimens (less than 7 mm) independently from the frequency and power employed. - Better quality of incision and less width of overall tissue injuries can be obtained with higher frequency and lower power - No statistical difference was observed between the two employed different parameters of the laser.
4. Suter et al., 2012 [22]	CO_2_	Excised benign exophytic lesions on the buccal mucosa (1 to 2 cm in diameter)	Fibrous hyperplasia (60)	60	Optical microscope	-	Histological evaluation including the measurement of the maximum width of the collateral thermal damage zone.	1	Similar medians of the histopathologic collateral damage zones were observed in the CW and PW groups.	- The CO2 laser can be safely used in both modes (CW and PW) for the excision of intraoral mucosal lesions. - For both modes, a 1 mm of safety margin is recommended.
5. Angiero et al., 2012 [28]	Diode 808 nm	Different oral soft tissue lesions (Excision for benign lesions or lesions ≤5 mm of diameter “performed retrospectively”)	Hyperkeratosis (72), fibroma (63), mucocele (73), lichen planus (64), squamous papilloma (53), hyperplasia (36), hemangioma (32), pyogenic granuloma (24), SCC (20), ulcer (17), lichenoid lesion (16), amalgam tattoo (14), traumatic neuroma (14), schwannoma (12), salivary gland neoplasia (10), peripheral giant cell granuloma (8), epulis fissuratum (7), peripheral ossifying fibroma (6), unreadable specimen (60)	608	Optical microscope	40× and 100×	Histological evaluation including epithelial, CT, vascular, cytological morphology changes, and thermal effect.	2	- Specimens larger than 3 mm were correctly diagnosed and did not show significant changes of all considered features. - Specimens under 3 mm showed frequently a significant epithelial, stromal and/or vascular changes where the diagnosis in 46.15% of these specimens was not achievable.	Diode laser (808 nm) can be safely used for the excision of oral lesions with a diameter larger than 3 mm.
6. Romeo et al., 2014 [32]	Diode 808 nm, KTP	Excised different oral soft tissue lesions (some lesions were incised because of site or size)	Fibroma (5), mucocele (3), hyperkeratotic lesion (4), oral lichen planus (3), giant cell granuloma (1), melanotic macula (1)	17	Optical microscope	5× and 10×	Histological assessment to evaluate quantitatively and qualitatively the marginal alterations.	1	- Histological analysis was not influenced for all the samples, and a certainty diagnosis was possible. - Due to the morphological and structural characteristics of the various lesions that may influence the tissues’ response to the laser, peripheral damage was individually evaluated for each disease.	- The histologic diagnosis was achievable by both lasers. - An extension of margins of about 0.5 mm of biopsies is suggested in particular in inflammatory lesions.
7. Gobbo et al., 2017 [33]	Diode 445 nm, 970 nm	Excised benign oral soft tissue lesions	Fibroma (35), angiofibroma (10), epulis (1), mucocele (8), papilloma (5), angioma (5), amalgam tattoo (1), papillae (1)	66	-	-	Histological evaluation and quantifying the maximum thermal damage along the cutting margin.	1	- The lowest thermal damage was observed in the blue laser group. - An obvious deference was observed between blue laser group and 970 nm laser group.	-All the tested instruments led to correct histological diagnosis. - Blue laser showed the minimal bleeding with limited thermal damage.
8. Suter et al., 2017 [23]	Er:YAG, CO_2_	Excised exophytic oral soft tissue lesions (with length and/or height from 5 mm to 20 mm)	Fibrous hyperplasia (31), non-identified lesion (1)	32	-	-	Histological evaluation and measurement of TDZ ^5^ (maximum and minimum).	1	Significant statistical difference was observed between CO_2_ laser group versus Er:YAG laser group in regard to the median of all minima, the median of all maxima, and the pooled median value (maxima and minima).	- The use of CO_2_ and Er:YAG lasers was valuable in excisional biopsy of oral soft tissues. - The Er:YAG laser showed the lower thermal effect which is considered an advantage for histopathological evaluation.
9. Monteiro et al., 2019 [34]	Diode 980 nm, Nd:YAG, Er:YAG, CO_2_	Excised benign oral soft tissue lesions (retrospectively)	Fibrous epithelial hyperplasia (95)	95	Optical microscope	5×, 20×, and 40×	Histological evaluation including epithelial, CT, vascular changes, incision morphology, and overall width of tissue modifications.	1	- The use of any lasers employed did not hinder the histological evaluation in all the samples. - Among the employed lasers, the highest tissue damage extension was observed with the diode laser, followed by Nd:YAG, CO_2_, and Er:YAG. - The most regular incision was observed with CO_2_ laser, followed by Er:YAG laser, Nd:YAG laser, and diode laser.	- Lasers can be used for the excision of benign lesions without histological limitations. - Er:YAG laser showed few tissue damage extension and good incision regularity, and can be the instrument of choice for excision of these lesions.
10. Suter et al., 2019 [24]	Er:YAG, CO_2_	Excised exophytic oral soft tissue lesions (with length and/or height from 5 mm to 20 mm)	Fibrous hyperplasia (49)	49	-	-	Histological evaluation including the measurement of the TDZ.	1	The Er:YAG group was significantly lower than CO_2_ laser group with regard to the median of all minimum values, the median of all maximum values, and the median of all values of the TDZ	The Er:YAG and CO_2_ lasers can be used for excisional biopsy of small lesions.
11. Tenore et al., 2019 [29]	CO_2_	Excised oral soft tissue lesions	Carcinoma in situ (1), mucocele (2), focal fibrous hyperplasia (4), kaposiform hemangioendothelioma (1), peripheral giant cell granuloma (1), granular cell tumor (1)	10	Optical microscope	100×	Histological evaluation including quantitatively and qualitatively assessing the thermal effect in both the epithelium and connective tissue.	1	- The thermal effects were limited to the surgical resection margins of all samples and did not hinder the histological diagnosis. - The thermal effect was greater in CT than that in epithelium in all samples except 1. - The most prominent thermal effect was observed on samples collected from attached gingiva.	Using CO_2_ laser for oral biopsy is valid and should be tailored with taking in consideration of different factors in addition to laser parameters.
12. Palaia et al., 2020 [30]	Blue diode (445 nm)	Excised benign soft tissue lesions (≤ 2 cm in diameter)	Focal fibrous hyperplasia (4), squamous papilloma (3), pyogenic granuloma (2), giant cell granuloma (1)	10	Optical microscope	100×	Histological evaluation including quantitative evaluation of the thermal effect at the peri-incisional margins for both the epithelium and CT.	1	- The resulted thermal effects did not hinder the histological diagnosis. - Almost all the cases showed thermal effect on both epithelium and CT lower than 1 mm except one case of focal fibrous hyperplasia with a thermal effect on epithelium more than 1 mm	The blue laser can be suggested as a safe tool for biopsies of clinically unsuspicious lesions.
13. Gill et al., 2021 [35]	Diode 808 nm, 980 nm	Oral soft tissue lesions including premalignant disorders (PMDs) (leucoplakia, erythroplakia and lichen planus)	-	70	Optical microscope	40×	Histological evaluation including quantitative evaluation of the LTD ^6^.	2	- Mean value of LTD in 808 nm group was five times that of 980 nm group. - The 808 nm has highest number of pathomorphological artefacts. - On histochemical evaluation, 80% of Giemsa-stained sections in scalpel group showed dense inflammation, 54.2% showed less dense in 808 nm group, and 34.2% sections in 980 nm group showed an absence of inflammation.	The 980 nm diode laser showed better results than the 808 nm.
14. Palaia et al., 2021 [31]	Blue diode (445 nm)	Excised benign oral soft tissue lesions	Focal fibrous hyperplasia (23), squamous papilloma (7), pyogenic granuloma (4), keratosis with no dysplasia (2), peripheral giant cell granuloma (2), lymphoepithelial cyst (1), mucocele (1), ectopic lymphoid tissue (1), granulation tissue (1)	42	Optical microscope	100×	Histological evaluation including quantitative evaluation of the thermal effect at the peri-incisional margins for both the epithelium and CT.	1	- The resulted thermal effects did not hinder the histological diagnosis. - The mean thermal effect extent was lower on CT than on epithelium. - The alteration extent was greater than 1 mm in only three specimens on the epithelium.	Blue diode laser has various intra- and postoperative advantages and is a safe tool for performing oral biopsy.

^1^ Squamous Cell Carcinoma (SCC); ^2^ Pulsed Wave (PW); ^3^ Continuous Wave (CW); ^4^ Connective Tissue (CT); ^5^ Thermal Damage Zone (TDZ), ^6^ Lateral Thermal Damage (LTD).

**Table 4 dentistry-11-00028-t004:** Overview of laser parameters and the resulted thermal effects of the included studies.

Author, Year	Study Type	Type of Emitter	Groups	Nr. of Samples (*n*)	Frequency/Energy	Air/H_2_O	Fiber/Spot Diameter (µm)	Fluence (J/cm^2^)	Irradiance (W/cm^2^)	Power (W)	Thermal Effects
Diode Laser (808 nm)											
1. Romeo et al., 2007 [10]	Animal	Diode	A	3	CW ^1^	-	320	2400	2400	2	3000 μm
			B	3	CW			1800	1800	1.5	1500 μm
			C	3	Ton 100 ms Toff 100 ms			248	2400	2	8–10 rows of keratinocytes (<1000 μm)
2. Angiero et al., 2012 [28]	Clinical	Diode	-	608	CW	-	320	-	-	1.6–2.7 (~2.5)	The width of thermal effect ranged from 260.7 μm to 321.4 μm, with an average of 282.8 μm.
3. Romeo et al., 2014 [32]	Clinical	Diode	-	15	CW	-	320	2400	-	2	In mucocele: 245 ± 162 μm. In fibroma: 382 ± 149 μm. In hyperkeratotic lesions: 336 ± 106 μm. In OLP ^2^: 473 ± 105 μm. In giant cell granuloma: 182 μm. In melanotic macula: 149 μm.
4. Fornaini et al., 2016 [17]	Animal	Diode	-	1	CW	-	320	-	-	3	Only the quality of cut score was reported and was 2.
5. Kawamura et al., 2019 [21]	Animal	Diode	-	8	CW	-	300	-	1415.4	1	The mean thickness of coagulated layer was 137.9 ± 19.5 μm, and the thermally affected layer was 123.1 ± 33.1 μm. The total thickness was 261.0 ± 45.1 μm.
6. Gill et al., 2021 [35]	Clinical	Diode	-	35	CW	-	-	-	-	2.5	The mean value of LTD ^3^ was 17.92 ± 12.495 μm
**Diode Laser (980 nm)**											
1. Romeo et al., 2007 [10]	Animal	Diode	D	3	CW	-	320	2400	2400	2	Collagen homogenization and dermoepithelial detachment.
			E	3	CW			1800	1800	1.5	>1000 μm in epithelium and >1500 μm in CT ^4^.
			F	3	Ton 100 ms Toff 100 ms			248	2400	2	Larger peripheral damage (collagen homogenization and dermo-epithelial detachment).
2. Azevedo et al., 2016 [13]	Animal	Diode	1	10	PW ^5^	-	-	-	-	3.5	The average ETTD ^6^ was 456.15 ± 108.513 μm.
			2	10	Boost PW		-	-	-	3.5	The average ETTD was 626.82 ± 220.292 μm.
3. Gobbo et al., 2017 [33]	Clinical	Diode (970 nm)	1	27	10 Hz	-	320	-	-	2	The mean thermal effect was 186.8 ± 82.7 μm.
4. Braun et al., 2018 [18]	Animal	Diode (970 nm)	1	-	CW	-	320 (contact mode)	-	-	3	Mean value of MDD ^7^ was 75 ± 41 μm. Mean value of DA ^8^ was 0.046 ± 0.043 mm^2^.
5. Monteiro et al., 2019 [34]	Clinical	Diode	1	21	50 Hz/70 mJ	-	300	99.2	4957.5	3.5	The mean epithelial damage distance was 913.73 ± 322.45 μm. The mean CT damage distance was 284.81 ± 110.56 μm.
6. Gill et al., 2021 [35]	Clinical	Diode	1	35	PW (50 ms interval)	-	-	-	-	5	The mean value of LTD was 3.808 ± 5.852 μm.
7. Palaia et al., 2021 [15]	Animal	Diode (976 nm)	A	5	CW	-	400	3184	-	4	The average epithelial peri-incisional effects was 0.2 ± 0.13 mm. The average CT peri-incisional effects was 0.3 ± 0.16 mm.
			B	5				3980		5	
			C	5				4777		6	
			D	5	PW (t_on_–t_off_: 100 ms–100 ms)			318.4		4	The average epithelial peri-incisional effects was 0.1 ± 0.14 mm. The average CT peri-incisional effects was 0.3 ± 0.13 mm.
			E	5				398		5	
			F	5				477.7		6	
8. Prado et al., 2022 [16]	Animal	Diode (940 nm)	G2	10	CW	-	300	-	1415.4	1	- Median TDA ^9^ was 1.09 mm^2^ - Median TDD ^10^ was 606.4 μm
			G3	10					2123.1	1.5	- Median TDA was 0.91 mm^2^ - Median TDD was 655.8 μm
			G4	10	PW (40 μs pulse interval) “CP0”				1415.4	1	- Median TDA was 1.22 mm^2^ - Median TDD was 720.4 μm
			G5	10					2123.1	1.5	- Median TDA was 1.37 mm^2^ - Median TDD was 789.6 μm
			G6	10	PW (200 μs pulse interval) “CP1”				1415.4	1	- Median TDA was 1.53 mm^2^ - Median TDD was 669.3 μm
			G7	10					2123.1	1.5	- Median TDA was 1.93 mm^2^ - Median TDD was 1024.9 μm
			G8	10	PW (1 ms pulse interval) “CP2”				1415.4	1	- Median TDA was 1.50 mm^2^ - Median TDD was 666.4 μm
			G9	10					2123.1	1.5	- Median TDA was 1.97 mm^2^ - Median TDD was 831.1 μm
			G10	10	Super PW (0.01 ms–20 ms pulse interval) “STP”		400		2547.7	3.2	- Median TDA was 1.08 mm^2^ - Median TDD was 778.8 μm
**Diode Laser (445 nm)**											
1. Fornaini et al., 2016 [17]	Animal	Blue diode (450 nm)	1	1	CW	-	320	-	-	2	Only the quality of cut score was reported and was 3.
2. Gobbo et al., 2017 [33]	Clinical	Blue diode	1	39	Ton 20 ms, Toff 8 ms	-	320	-	-	1.4	The mean thermal effect was 71.3 ± 51.8 μm.
3. Braun et al., 2018 [18]	Animal	Blue diode	1	-	CW	-	320 (contact mode)	-	-	2	Mean value of MDD was 59 ± 18 μm. Mean value of DA was 0.050 ± 0.034 mm^2^.
			2	-			320 (non-contact mode (1 mm))	-	-	2	Mean value of MDD was 65 ± 27 μm. Mean value of DA was 0.039 ± 0.017 mm^2^.
4. Palaia et al., 2020 [14]	Animal	Diode	A	5	CW	-	320	-	-	2	Epithelium was 137.5 ± 74.28 μm, CT was 416.25 ± 118.73 μm.
			B	5	CW			-	-	3	Epithelium was 145.6 ± 101.88 μm, CT was 546.2 ± 359.49 μm
			C	5	CW			-	-	4	Epithelium was 85.2 ± 64.27 μm, CT was 321.6 ± 132.51 μm.
			D	5	50 Hz			-	-	2	Epithelium was 69 ± 34 μm, CT was 328 ± 169.12 μm.
			E	5	50 Hz			-	-	3	Epithelium was 80.2 ± 31 μm, CT was 250.6 ± 144.16 μm.
			F	5	50 Hz			-	-	4	Epithelium was 55.8 ± 33.51 μm, CT was 320.8 ± 191.82 μm.
5. Palaia et al., 2020 [30]	Clinical	Diode	1	10	CW	-	320	3100	-	2.5	The average thermal effect was 650.93 ±311.96 μm on epithelium, and 468.07 ± 264.23 μm on CT.
6. Palaia et al., 2021 [31]	Clinical	Diode	1	42	CW	-	320	3100	-	2.5	The mean thermal effect was 507.08 ± 283.76 µm on epithelium, and 320.39 ± 209.04 µm on CT.
**Nd:YAG Laser (1064 nm)**											
1. Romeo et al., 2007 [10]	Animal	Nd:YAG	G	3	40 Hz/120 mJ	-	400	95.5	3800	4.8	Wide epithelium detachment and basal layer damage.
			H	3	50 Hz/120 mJ			95.5	4700	6	At least 1500 μm of epithelial damage, and involving CT damage.
			I	3	90 Hz/60 mJ			47.7	4300	5.4	The most damaged specimens, one of the specimens was so compromised with only <0.5 cm of the specimen was interpretable.
2. Vescovi et al., 2010 [26]	Clinical	Nd:YAG	1	6	60 Hz with pulse width of 100 µs	-	320	-	488.281	3.5	305.8 μm of epithelial changes, 376.6 μm of CT changes, and 151.6 μm of vascular changes.
			2	9	30 Hz with pulse width of 100 µs			-	300.000	5	399.8 μm of epithelial changes, 521 μm of CT changes, and 183.5 μm of vascular changes.
3. Azevedo et al., 2016 [13]	Animal	Nd:YAG	1	10	40 Hz	-	300	-	-	6	The average ETTD was 670.68 ± 251.85 μm.
4. Fornaini et al., 2016 [17]	Animal	Nd:YAG	1	1	30 Hz	-	320	-	-	3	Only the quality of cut score was reported and was 1.
5. Kawamura et al., 2019 [21]	Animal	Nd:YAG	1	8	20 Hz (with pulse duration of 100 µs)/50 mJ	-	300	70.8 per pulse	-	1	The mean thickness of coagulated layer was 137.8 ± 16.3 μm, and the thermally affected layer was 161.8 ± 41.2 μm. The total thickness was 299.6 ± 49.6 μm.
6. Monteiro et al., 2019 [34]	Clinical	Nd:YAG	1	25	40 Hz/100 mJ	-	300	141.6	5665.7	4	The mean epithelial damage distance was 899.83 ± 327.75 μm. The mean CT damage distance was 310.85 ± 107.45 μm.
**Er,Cr:YSGG Laser (2780 nm)**											
1. Rizoiu et al., 1996 [19]	Animal	Er,Cr:YSGG	-	1 (Not clear)	20 Hz with pulse duration of 140 µs	-	680	-	-	2	The margins showed a 20–40 μm of carbonization and coagulation.
2. Romeo et al., 2007 [10]	Animal	Er,Cr:YSGG	L	3	20 Hz	11/07	600	44.2	884	2.5	Keratinocytes degeneration for 1500 μm.
			M	3		11/10		35	707	2	Dermo-epithelial detachment for 2500 μm.
			N	3		11/15		53	1000	3	1000 μm in both epithelium and CT.
3. Cercadillo-Ibarguren et al., 2010 [3]	Animal	Er,Cr:YSGG	1	9	20 Hz	11/07	600	-	-	1	The mean thermal effect was 9.26 ± 2.05 μm.
			2	9		without		-	-	1	The mean thermal effect was 23.38 ± 5.31 μm.
			3	9		11/07		-	-	2	The mean thermal effect was 14.88 ± 4.0 9 μm.
			4	9		without		-	-	2	The mean thermal effect was 55.67 ± 17.65 μm.
			5	9		11/07		-	-	4	The mean thermal effect was 13.42 ± 5.61 μm.
4. Kawamura et al., 2019 [21]	Animal	Er,Cr:YSGG	1	8	20 Hz (with pulse duration of 140 µs)/50 mJ	without	600	17.7 per pulse	-	1	The mean thickness of coagulated layer was 50.6 ± 6.7 μm, and the thermally affected layer was 65.7 ± 27.6 μm. The total thickness was 116.4 ± 31.7 μm.
			2	8		with			-	1	The mean thickness of coagulated layer was 33.1 ± 1.4 μm, and the thermally affected layer was 0 μm. The total thickness was 33.1 ± 1.4 μm.
**KTP Laser (532 nm)**											
1. Romeo et al., 2010 [11]	Animal	KTP	1	9	Ton 50 ms Toff 50 ms	-	300	141	2800	2	Limited to only 2–3 lines of cells (<50 μm).
			2	9	Ton 50 ms Toff 50 ms			162	3200	2.3	Signs of poor tissue coagulation both in the epithelium and CT (<50 μm).
			3	9	Ton 50 ms Toff 50 ms			176	3500	2.5	Highest amount of thermal damage over the epithelium, but the subepithelial corium was absolutely free from thermal alteration.
			4	9	Ton 50 ms Toff 50 ms			191	3800	2.7	Few signs of cellular hyperthermia near the cut edges, but they were always readable and showed clear details.
			5	9	Ton 50 ms Toff 50 ms			212	4200	3	Almost intact epithelium and signs of cellular damage were limited to 200 μm.
2. Romeo et al., 2014 [32]	Clinical	KTP	-	2	PW	-	300	212	-	1.5	In mucocele: 213 μm. In hyperkeratotic lesion: 196 μm.
3. Fornaini et al., 2016 [17]	Animal	KTP	1	1	CW	-	320	-	-	2	Only the quality of cut score was reported and was 3.
**Er:YAG (2940 nm)**											
1. Romeo et al., 2012 [12]	Animal	Er:YAG	1	9	30 Hz/60 mJ	Not used	600	21	-	1.8	Little peripheral thermal damage with clear and safe margins, with modifications confined to the lamina propria and muscle tissue (TDS ^11^ 1.44).
			2	9	30 Hz/80 mJ			28	-	2.4	A very thin layer of cauterization (TDS 1.1).
			3	9	30 Hz/100 mJ			35	-	3	Limited signs of thermal damage in both epithelium lamina propria, although the subepithelial chorion was free from thermal alteration (TDS 1.22).
			4	9	30 Hz/130 mJ			46	-	3.9	Moderate damage localized to the epithelial margins (TDS 1.77).
			5	9	30 Hz/150 mJ			53	-	4.5	Significant signs of peripheral thermal damage in the epithelium (TDS 2.44).
2. Azevedo et al., 2016 [13]	Animal	Er:YAG	1	10	10 Hz/0.2 J (short pulse)	With	Not mentioned (non-contact)	-	-	2	The average ETTD was 68.39 ± 59.585 μm.
			2	10	10 Hz/0.2 J (short pulse)	Without		-	-	2	The average ETTD was 84.39 ± 51.363 μm.
			3	10	10 Hz/0.4 J (short pulse)	With		-	-	4	The average ETTD was 66.34 ± 25.143 μm.
			4	10	10 Hz/0.4 J (short pulse)	Without		-	-	4	The average ETTD was 79.54 ± 31.333 μm.
3. Suter et al., 2017 [23]	Clinical	Er:YAG	1	16	35 Hz (pulse duration of 297 μs)/ 200 mJ	Air-water cooling (22.5 mL/min)	400 (non-contact)	-	-	7	The median of all TDZ values was 34.0 μm. The median of all maxima TDZ was 56.8 μm. The median of all minima TDZ was 18.2 μm.
4. Kawamura et al., 2019 [14]	Animal	Er:YAG	1	8	20 Hz (with pulse duration of 200 µs)/50 mJ	without	600	17.7 per pulse	-	1	The mean thickness of coagulated layer was 37.7 ± 9.6 μm, and the thermally affected layer was 0 μm. The total thickness was 37.7 ± 9.6 μm.
			2	8		with			-	1	The mean thickness of coagulated layer was 17.9 ± 2.1 μm, and the thermally affected layer was 0 μm. The total thickness was 17.9 ± 2.1 μm.
5. Monteiro et al., 2019 [34]	Clinical	Er:YAG	1	22	20 Hz/200 mJ	-	500	102	2040.8	4	The mean epithelial damage distance was 166.47 ± 123.85 μm. The mean CT damage distance was 48.54 ± 26.09 μm.
6. Suter et al., 2019 [24]	Clinical	Er:YAG	1	25	35 Hz (with pulse duration of 297 µs)/200 mJ	22.5 mL/min	400 (non-contact)	-	-	7	The mean TDZ ^12^ was 41 μm.
**CO_2_ Laser (10,600 nm)**											
1. Pogrel et al., 1990 [27]	Clinical	CO_2_	1	23	-	-	-	-	2320	17.5	Mean width of necrosis in epithelium was 85.9 ± 15.7 μm. Mean width of necrosis in muscle was 85.1 ± 12.8 μm. Mean width of necrosis in dense CT was 96.1 ± 22 μm. Mean width of necrosis in loose CT was 51.1 ± 14.9 μm. Mean width of necrosis in salivary gland was 41.5 ± 8 μm.
2. Matsumoto et al., 2008 [25]	Clinical	CO_2_	1	10	PW	-	1000	-	-	3–4	The range of thermal artifacts was from 210–320 μm, with lowest range of 269 ± 38.72 μm.
			2	10	CW			-	-	3–4	The range of thermal artifacts was from 240–360 μm with lowest range of 306 ± 32.04 μm.
3. Cercadillo-Ibarguren et al., 2010 [3]	Animal	CO_2_	1	9	CW	-	-	-	-	1	The mean thermal effect was 21.55 ± 8.24 μm.
			2	9	CW			-	-	2	The mean thermal effect was 35.16 ± 15.93 μm.
			3	9	CW			-	-	10	The mean thermal effect was 20.44 ± 5.43 μm.
			4	9	CW			-	-	20	The mean thermal effect was 29.02 ± 13.56 μm.
			5	9	PW (with 100 msec pulse width and 200 msec interval)			-	-	20	The mean thermal effect was 20.30 ± 6.73 μm.
4. Seoane et al., 2010 [20]	Animal	CO_2_	1	5	PW (0.05 s)	-	-	-	-	3	The mean width of thermal damage was 330 ± 237.7 μm.
			2	5				-	-	6	The mean width of thermal damage was 300 ± 79.0 μm.
			3	5				-	-	9	The mean width of thermal damage was 320 ± 119.1 μm.
			4	5				-	-	12	The mean width of thermal damage was 245 ± 158.5 μm.
5. Suter et al., 2012 [22]	Clinical	CO_2_	1	30	CW	-	200	-	-	5	The mean of width of the collateral thermal damage zone was 247.2 μm.
			2	30	140 Hz (pulse duration of 400 μs)/ 33 mJ			-	-	4.62	The mean of width of the collateral thermal damage zone was 198.4 μm.
6. Palaia et al., 2014 [5]	Animal	CO_2_	A	5	CW	-	200/400	-	-	2	Epithelium was 99 μm, CT was 169 μm.
			B	5	CW			-	-	3	Epithelium was 68 μm, CT was 184 μm.
			C	5	CW			-	-	4	Epithelium was 90 μm, CT was 239 μm.
			D	5	50 Hz			-	-	3	Epithelium was 74 μm, CT was 192 μm.
			E	5	50 Hz			-	-	3.5	Epithelium was 77 μm, CT was 206 μm.
			F	5	50 Hz			-	-	4	Epithelium was 71 μm, CT was 236 μm.
7. Azevedo et al., 2016 [13]	Animal	CO_2_	1	10	50 Hz	-	Not mentioned (non-contact)	-	-	3.5	The average ETTD was 306.19 ± 85.882 μm.
			2	10	50 Hz			-	-	7	The average ETTD was 485.45 ± 178.581 μm.
			3	10	CW			-	-	7	The average ETTD was 571.18 ± 183.216 μm.
8. Suter et al., 2017 [23]	Clinical	CO_2_	1	15	140 Hz (pulse duration of 400 μs)/ 33 mJ	-	200 (non-contact)	-	-	4.62	The median of all TDZ values was 74.9 μm. The median of all maxima TDZ was 122.6 μm. The median of all minima TDZ was 49.9 μm.
9. Kawamura et al., 2019 [21]	Animal	CO_2_	1	8	CW	-	400 (non-contact)	-	796.2	1	The mean thickness of coagulated layer was 163 ± 48.8 μm, and the thermally affected layer was 73.6 ± 23.4 μm. The total thickness was 236.6 ± 50.2 μm.
10. Monteiro et al., 2019 [34]	Clinical	CO_2_	1	27	80 Hz/50 mJ	-	500	40.8	2040.8	4	The mean epithelial damage distance was 538.37 ± 170.50 μm. The mean CT damage distance was 201.69 ± 89.86 μm.
11. Suter et al., 2019 [24]	Clinical	CO_2_	1	24	140 Hz (pulse duration of 400 μs)/33 mJ	with air cooling	200 (non-contact)	-	-	4.62	The mean TDZ was 83.5 μm.
12. Tenore et al., 2019 [29]	Clinical	CO_2_	1	10	80 Hz	-	200–400 (non-contact)	-	-	4.2	The average thermal effect was 687 μm on epithelium, and 1470 μm on CT. The average total thermal effect was 2094 μm.

^1^ Continuous Wave (CW); ^2^ Oral Lichen Planus (OLP); ^3^ Lateral Thermal Damage (LTD); ^4^ Connective Tissue (CT); ^5^ Pulsed Wave (PW); ^6^ Extent of Thermal Tissue Damage (ETTD); ^7^ Maximum Denaturation Depth (MDD); ^8^ Denaturation Area (DA); ^9^ Thermal Damage Area (TDA); ^10^ Thermal Damage Depth (TDD); ^11^ Thermal Damage Score (TDS); ^12^ Thermal Damage Zone (TDZ).

## Data Availability

Not applicable.

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
