# Peer review of "The Impact of Laser Thermal Effect on Histological Evaluation of Oral Soft Tissue Biopsy: Systematic Review"

_dentistry, 2023, doi:10.3390/dj11020028_

Round 1
Reviewer 1 Report
In this systematic review, Tenore and colleagues found that almost all the studies indicated that the thermal effect of different lasers did not hinder the histological diagnosis and evaluation. Moreover, thermal damage may occur with different wavelengths and parameters, which can be minimized by adjusting laser parameters. Last but not least, the extension of margins during the collection of laser oral biopsies was recommended, while laser should only be used in non-suspicious lesions. This work is highly clinically relevant especially for oral pathologists. I only have a few comments:
1. Line 18-20, please expand several abbreviations, including Nd:YAG, CO2, KTP, and Er:Cr:YSGG, as these were the first time when those terms were introduced in the Abstract.
2. Was this systematic review registered on PROSPERO? If yes, the serial number should be listed. If not, please register this work on PROSPERO and provide the serial number.
3. Line 51-53, please expand several abbreviations, including KTP, Nd:YAG, Er:Cr:YSGG, Er:YAG laser, and CO2, as these were the first time when those terms were introduced in the text. In particular, all abbreviations should be expanded when they are introduced in the text.
4. In "2.2. Search Strategy", please list the entire Boolean logic containing all applied search terms used for search syntax.
5. Line 21 and Line 241, please replace "didn't" with "did not". Likewise, please revise all these kinds of use throughout the manuscript.
6. Line 347-364, the conclusions were too lengthy. Please make it more concise into no more than 4 sentences, and move the rest contents into Discussion.
7. For Table 2, what were the primary diagnoses, and how was the clinicopathological correlation? These are important components from a pathological perspective which should be listed in the table.
I look forward to seeing a revised version of this work!
Reviewer 2 Report
Thank you for submitting "The impact of Laser Thermal Effect on Histological Evaluation of Oral Soft Tissue Biopsy: Systematic Review"
The aim of the study is to review the literature to observe studies assessing the thermal effect with different laser wavelength r on the histological evaluation of oral soft tissue biopsies and to analyse the difference on the resulted thermal effects in soft tissue specimens collected by different wavelengths and parameters.
Abstract: add when the search was done. Add the risk of bias of the selected studies.
Introduction: it is too short.
Mat and method: Was the systematic review registered on Prospero?
Search strategy: Which were the MeSH term? I want to do the search just like you did. Put all the words we need to duplicate the search.
Round 2
Reviewer 2 Report
The authors have followed the changes suggested and have greatly improved the article. Therefore, in my opinion, this scientific article meets the necessary criteria to be published in present form.
Author Response
Thank you for the valuable revision and comment. We believe that the article is improved after performing your requested corrections and modifications.